# Bony Canal Method of Dexamethasone Injections in Aggressive Form of Central Giant Cell Granuloma—Case Series

**DOI:** 10.3390/medicina59020250

**Published:** 2023-01-28

**Authors:** Iwona Niedzielska, Mateusz Bielecki, Michał Bąk, Barbara Dziuk, Damian Niedzielski

**Affiliations:** Department of Cranio-Maxillo-Facial Surgery, Medical University of Silesia in Katowice, ul. Francuska 20-24, 40-027 Katowice, Poland

**Keywords:** Central Giant Cell Granuloma (CGCG), Central Giant Cell Lesion (CGCL), dexamethasone, oral tumor

## Abstract

Central Giant Cell Granuloma constitutes approximately 7% of benign tumors of the jaws. The aggressive form of CGCG clinically behaves like a classic semi-malignant neoplasm. In the literature, the suggested method of treatment of aggressive forms of CGCG is curettage or resection with the margin of 0.5 cm. Surgical treatment, especially in the developmental age, entails disturbances in the growth and differentiation of tissues and deforms and disturbs the functioning of the stomatognathic system. Alternative treatment methods of the CGCG presented in this article lead to the patient avoiding a mutilating procedure and improve their quality of life. The aim was to present alternative method of treatment of aggressive forms of Central Giant Cell Lesion of the jaws—injections of dexamethasone into the tumor mass through drilled bony canals. Here, we present the three cases of aggressive forms of CGCG of jaws treated with dexamethasone injections into the tumor mass. Two cases resulted in regression of the tumor, which was confirmed in histologic evaluation after remodeling surgery. Those two patients were uneventful and showed no signs of tumor recurrence at 8 and 9 years of thorough follow-up, respectively. The third patient was qualified for the mandible resection due to the enlargement of the lesion and destruction of the cortical bone. According to our observations, if the proper patient discipline, and thorough, careful clinical and radiological examinations are provided, the dexamethasone injections could be a recommended method of treatment of intraosseous giant cell granuloma. The indication is restricted to the cases with preserved bony borders despite deformation. Additionally, leaving vital teeth in the lesion is also possible.

## 1. Introduction

The endosseous giant cell lesion, Central Giant Cell Granuloma (CGCG), is a benign lesion developing as a bone tissue response to an injury, inflammation or as a reparative bone reaction [1,2]. CGCG constitutes approximately 7% of benign tumors of the jaws. It is usually located in the facial skeleton, mainly in the mandible (75%) and the maxilla [3,4]. The lesion usually is unifocal, although the cases of bifocal or multifocal location have been described, particularly in patients with Noonan syndrome [5], neurofibromatosis type I, Paget’s disease, and cherubism [6,7]. It occurs in patients of different ages. It is usually detected in the second or third decade of life, more often in women, therefore some authors link female hormones secretion with the development of the disease [8,9]. The lesion may also occur in patients at the developmental age, more often in boys [5].

The clinical features of the tumor vary. Two forms of CGCG can be distinguished. The non-aggressive form (60–80%) is characterized by slow growth, low severity of symptoms, no cortical bone destruction and root resorption, and low likelihood of recurrence [10,11,12,13]. The second one is the aggressive form (19–40%), which clinically behaves like a classic semi-malignant neoplasm and is characterized by rapid growth, localized pain, progressing facial asymmetry, root resorption, paraesthesias, destruction of the cortical bone, pathological fractures, and frequent recurrences [13,14,15,16,17,18]. The radiological image of CGCG, according to many authors [10,19,20,21,22], presents as an osteolytic bone defect with fairly distinct boundaries. The non-aggressive form is usually single cavity without features of teeth displacement and root resorption. Tumor area often has small, round bone calcifications that can form septa. If the septa exit the periphery of the focus at right angles and cause incisions in the stretched outer cortical plate of the bone, this is the image characteristic of CGCG. In the aggressive form, an osteolytic multi-cavity lesion with the septa is observed, causing significant deformation of jaw. Teeth displacement [19,20] as well as root resorption [20,21] frequently occur. It should be noted that the radiological picture is not pathognomonic for CGCG and may suggest a different lesion. CGCG also shows no pathognomonic histological changes. Histological analysis reveals giant cells around a fibrous stroma and a degree of cellular atypia and is very similar to the brown bone tumor found in hyperparathyroidism [22,23,24,25,26].

Therefore, blood test results should be considered in the differential diagnosis to rule out a brown tumor. An increased level of calcium, alkaline phosphatase and parathormone in the serum or a decreased level of phosphorus indicate hyperparathyroidism. Radiological imaging should be differentiated from single bone cysts, odontogenic cysts and granulomas (non-aggressive form), aneurysmal cyst, solid ameloblastoma and odontogenic myxoma (the aggressive form) [8,27,28].

According to the literature, the treatment of CGCG depends on the location, extent and aggressiveness of the lesion and patient’s age [16,21,26]. The traditional treatment of CGCG is surgical curettage of the lesion or resection of the involved bone section with the 0.5 cm margin [14,16,26,27]. Chuong et al. claim that the large size of the lesion, as well as its aggressive character, qualify it exclusively for surgical treatment [28]. Surgical treatment, especially in the developmental age, entails disturbances in the growth and differentiation of tissues as well as deforms and disturbs the functioning of the stomatognathic system.

For this reason, alternative methods for CGCG treatment are being sought (corticosteroids, calcitonin, interferon alpha, radiation therapy). However, isolated scientific reports of Body et al. in 1981 [29] and Terry and Jacoway in 1988 [30], which concerned pharmacological methods for small and non-aggressive forms of CGCG, failed to implement similar methods for aggressive forms. The authors suggested to apply local steroid therapy for at least 6 weeks (usually, dexamethasone was used once a week) and ending treatment with surgery after lesion reduction [6,11,16,20]. The main disadvantages of this method mentioned by the authors were the long treatment period, the need for frequent check-up visits, and the need for patient’s strong discipline [22,31,32,33].

Another pharmacological treatment method for non-aggressive forms of CGCG recommended by some authors include the use of calcitonin intranasally or by subcutaneous injection [26,27,34,35,36]. The function of calcitonin is to inhibit the activity of osteoclasts and thus limit the destruction of bone tissue. The treatment is considered effective, but the long duration of therapy and possible complications limit its use [25,37].

Other authors have suggested the use of interferon alpha-2a in the treatment of CGCG. However, due to its significant side effects, it should only be used in cases where other methods have failed or as a complement to surgical treatment [8,34,35].

In the available literature, no reports of non-surgical treatment of CGCG aggressive forms in developmental age were detected.

The Department of Maxillofacial Surgery treated a large number of CGCG cases between 2011 and 2022, with each case being treated individually.

It seems important to develop a new strategy for treating the aggressive form of CGCG. Treatment of those lesions, according to current guidelines requiring resection surgery, involves significant trauma to the patient, the loss of many teeth, and the need for a complex reconstructive procedure, even a free fibula flap surgery. This, especially for young patients, during the growth period, can significantly complicate their facial development, including at the recipient site.

The purpose of this study was to clinically, radiologically and histopathologically evaluate the efficacy of a novel method of intraoperative administration of dexamethasone through bony canals to the tumor mass as a sole and definitive treatment option in cases of extensive, aggressive giant cell tumors with long-term follow-up.

## 2. Case Series

The study included three patients with an extensive aggressive form of CGCG confirmed by histopathological examination. Patients were qualified for a nonsurgical treatment—dexamethasone injection into tumor mass—based on the history, clinical and radiological examinations.

Three patients reported to our Maxillofacial Surgery Department between 2013 and 2017. In all cases, medical history did not include systemic burdens, genetic load, or an injury from the past, except hypothyroidism in one patient. All the patients did not present any abnormalities in a physical examination except lesions in jaws. In addition, laboratory tests, including the level of parathormone (PTH), calcium and phosphates were correct.

Due to the large size of the tumors, young age, and good communication with the patients and their families guaranteeing the discipline of follow-up visits, surgery was abandoned and the patients were qualified for nonsurgical, pharmacological treatment. It was carried out by a series of dexamethasone (Dexaven, Jelfa, Jelenia Góra, Poland) injections into the tumor mass after drilling of two openings in the jawbone at the top of the bone deformities that corresponded to the largest bone defects in the CBCT studies. All the patients remained under permanent control of the Maxillofacial Surgery Outpatient Department.

### 2.1. Case #1

Patient M.K., male, aged 14 reported due to a tumor of the body of the mandible diagnosed in February 2013. The patient was obese. Physical examination revealed facial asymmetry with the distention of the body of the mandible on the right side at teeth 41–47 with decrease in the depth of the oral vestibule and the elevation of the floor of the oral cavity in the areas of both sublingual folds (Figure 1). The consistency of the distension was firm.

An orthopantomographic radiograph showed a multi-cavity osteolytic defect with septa extending from tooth 33 to 47 region (Figure 2a). CT scan revealed “the presence of nodular remodeling of the body of the mandible from the incisor on the left side to first molar on the right side. The tumor causes distension of the mandible and destruction of the cortical layer and diploe”. The tumor dimensions were 37 × 40 × 33mm (AP × SD × CC). In addition to lesions involving the mandible, pathological masses caused deformation of soft tissues (Figure 2b).

From April to June 2013, the patient was administered 12 mg/3 mL of Dexaven to the tumor mass every 7 days, and then for the next 3 months 8 mg/2 ml of Dexaven was administered every 14 days (Figure 1). The injections were administered through the bony canals, which were created by the means of surgical drill (2 mm diameter) under local anesthesia, especially for that purpose. Tumor structure and bone tissue recovery were assessed using control orthopantomographic radiographs. During treatment, headaches and an increase in blood pressure occurred. A follow-up CT scan was performed, which showed no abnormalities in the neurocranium. The patient was diagnosed with hypertension and treated with pharmacological therapy. The patient underwent a follow-up CBCT scan of the mandible 6 months later, which showed normal progressive bone remodeling. In October 2013 (Figure 3a,b), the patient underwent a mandibular body modeling surgery procedure using piezo surgery with macroscopic evaluation of bone tissue, mainly the cancellous bone, and collection of material for histopathological examination. Histopathological examination revealed trabecular bone tissue without atypia.

Currently, 9 years after the completion of therapy and mandibular body modeling surgery, clinically and radiologically, there are no signs of recurrence (Figure 4a,b).

### 2.2. Case #2

Patient A.S., female, aged 29 reported to our department due to a tumor of the hard palate. The lesion appeared for the first time in June 2012 during pregnancy, and at the time when pregnancy was lost, it subsided. It appeared again during the second pregnancy in October 2012. In January 2014, the resection of tooth 12 root was performed, resulting in a short-term reduction in the lesion. Physical examination revealed firm palatal distension in the midline. The CBCT showed a multi-cavity osteolytic defect in the maxilla in region of tooth 13 to 24 (Figure 5a,b).

In July 2014, a tumor specimen was collected. Histopathological examination revealed the diagnosis of “Tumor gigantocellularis pigmentosus”. From September to December 2014, the patient was administered, through two bony canals into the tumor mass, 12 mg/3 mL of Dexaven every 7 days, followed by 8 mg/2 mL of Dexaven every 14 days for the next 3 months. Tumor structure and bone tissue recovery were assessed using control orthopantomographic radiographs. The patient underwent a follow-up CBCT scan of the mandible 4 months after the start of therapy, which showed normal progressive bone remodeling. In February 2015, the patient underwent modeling surgery of the deformed hard palate using piezo surgery with macroscopic evaluation of bone tissue and collection of material for histopathological examination (Figure 6).

Macroscopically, normal cancellous bone was identified and a small focus of altered tissues on the palate in the midline. The material for histopathological examination was collected in three specimens. Histopathological examination revealed tiny fragments of connective tissue and bone fragments and tumor fragments composed of spindle cells and osteoblastic giant cells with visible deposits of hemosiderin, and no histological features of giant-cell tumor. Currently, after 8 years of therapy and hard palate modeling surgery, there are no clinical or radiological signs of recurrence (Figure 7).

### 2.3. Case #3

Patient M.K., female, aged 18 reported due to a tumor of the body of mandible which was detected in January 2017. The onset of complaints in the form of numbness of the lower lip was diagnosed at the dental office and subsequently treated neurologically without effects. In April 2017, there was swelling of the mandibular body—a orthopantomographic radiograph was collected, which showed an osteolytic lesion of the body of the mandible. A diagnostic biopsy was performed and showed “a lesion consisting of fields of fibroblasts with hemorrhages and numerous multinucleated giant cells and blood-filled spaces.” At this time, physical examination revealed asymmetry with swelling of the chin and mandibular body areas—excessively hard, painful, reddened skin over the tumor, bilateral hypoesthesia of the third branch of the trigeminal nerve, loosening of teeth 33–43, reddening of the adjacent mucosa and a bluish-purple tumor with elastic consistency. A diagnostic biopsy was performed that showed “a lesion consisting of fields of fibroblasts with hemorrhages and numerous multinucleated giant cells and blood-filled spaces.” In June 2017, a tumor specimen was collected for histopathological examination revealing the CGCG. From June to August 2017, the patient was administered a total of 84 mg/21 mL of Dexaven to the tumor mass through two bone canals in divided doses at 2-day intervals while being advised to undergo endodontic treatment of teeth 33–43 and 70 mg of alendronic acid orally once a week. Tumor hardness and structure were continuously monitored during follow-up examinations performed on injection days. In September 2017, due to the enlargement of the lesion and destruction of the cortical bone found on clinical examination and follow-up radiograph, as well as the onset of steroid acne, drug treatment was discontinued and surgical segmental resection of the mandibular body from the area of tooth 37 to the angle on the right side with implantation of a reconstructive plate was performed.

The patient was then qualified for reconstructive surgery using a free fibula flap (FFF) collected from the left lower extremity with simultaneous placement of four dental implants; the surgery was performed in October 2017. Currently, 5 years after the end of therapy, there are no clinical of radiological signs of recurrence. The summary of basic data for the case series is presented in Table 1.

## 3. Discussion

CGCG usually occurs in patients in the 2nd and 3rd decades of life, so the age of the patients presented in this article corresponds to the data presented. In the literature, aggressive forms of CGCG that model the surrounding soft tissues are detected more often in the mandible than in the maxilla.

Based on clinical features, there are two forms of CGCG: a non-aggressive form, characterized by slow growth and scanty symptoms, and an aggressive form, characterized by rapid growth, pain, bone destruction, displacement of teeth or their embryos, resorption of tooth roots [10,11]. According to the literature, both types require radical treatment, i.e., segmental resection with reconstruction. One of the authors proved that giant cell epulis more closely resembles oral squamous cell carcinoma than other types of epulis [24]. Since the origin of the CGCG and the role and number of giant cells is still unclear, various therapies are being considered, the efficacy of which is under constant review. The literature admits alternative treatments to surgery, such as triamcinolone injections, dexamethasone, the use of bisphosphonates, calcitonin, and interferon alfa-2a.

What is missing in the available research is the evaluation of the dentition with the preservation of viable teeth and a final histopathological assessment of therapeutic success. We found no case reports in the literature in terms of the exact size of the tumor, the viability of the teeth and definitive verification by histopathological examination after nonoperative treatment. There is also no description of the technique of depositing drugs, including dexamethasone, into the calcified tumor mass. In the cases we described, tooth viability was preserved after treatment. Surgery was required to reshape the bone and collect material for histopathological verification. Leaving viable teeth in the tumor is questioned in the literature, but the authors of the study proved the correctness of leaving viable teeth provided regular examinations, which may suggest the possibility of such a procedure in patients with discipline.

The third case presented seems to have been unsuccessful due to the lack of bone constraints on the tumor, and thus insufficient healthy bone tissue around the tumor, which is a source of osteoblastic cells necessary for the regeneration process. The surrounding bone was thin and the teeth were loosened. Although a follow-up orthopantomographic image showed the formation of subtle trabecular tissue inside the tumor, we decided to perform radical surgery.

Most authors take the view that surgical removal is the only treatment option for large and aggressive forms of CGCG. This type of treatment is unfortunately associated with post-surgical facial deformities and loss of teeth or their embryos, and thus the need for reconstructive procedures such as autogenous bone grafting, which carries the risk of serious complications, especially in children [21,23,38].

Another point worth considering is that even after surgery, aggressive forms tend to recur in 37.5% of cases. In the clinical cases described in this article, due to the large size of the tumors, the immature age of the patients, and good communication to ensure the discipline of follow-up visits, surgery was abandoned in two of the three cases and the patients were qualified for series of dexamethasone injections into the tumor mass. Most of the literature allows pharmacological treatment only for lesions of minor severity and non-aggressive form.

In two of the described cases of aggressive, large lesions, pharmacotherapy was fully successful, while in the third case, due to complication (occurrence of steroid acne) and failure to achieve full therapeutic effect, surgical treatment was performed. The likely mechanism of action of dexamethasone injected into the tumor mass is to inhibit the production of lysosomal proteases involved in bone resorption by giant cell tumors, as well as induce osteoclast apoptosis, thereby accelerating bone regeneration.

There are no reports in the available literature on pulp vitality and root canal treatment of teeth embedded in tumor masses. In the two described cases of successful pharmacological treatment, endodontic treatment of teeth embedded in the tumor mass was abandoned due to the absence of tooth root resorption and normal pulp response to thermal vitality tests.

## 4. Conclusions

Dexamethasone injections could be the recommended treatment for intraosseous giant cell tumors in cases with preserved bony borders despite the deformity, with teeth left in the lesion if their pulp is vital during treatment. Further investigation is needed in order to create qualification criteria and clinical recommendations.

Therapeutic success depends on patient discipline and regular injections two to three times a week without interruption for at least three months with regular clinical and radiological follow-up and monitoring for complications.

## Figures and Tables

**Figure 1 medicina-59-00250-f001:**
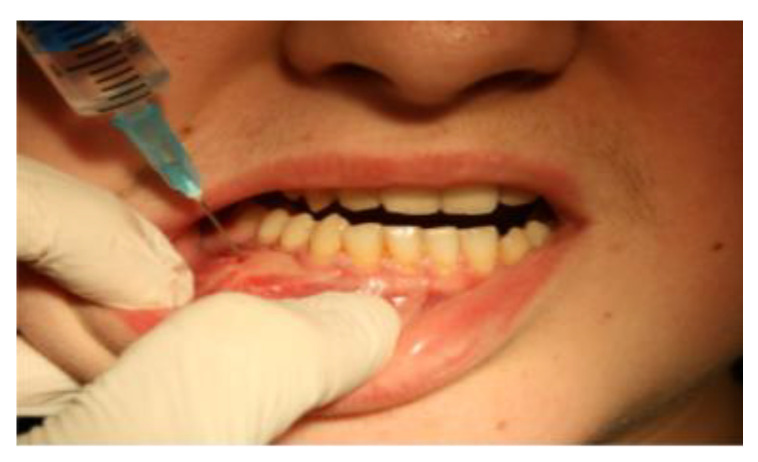
Case #1 initial clinical presentation.

**Figure 2 medicina-59-00250-f002:**
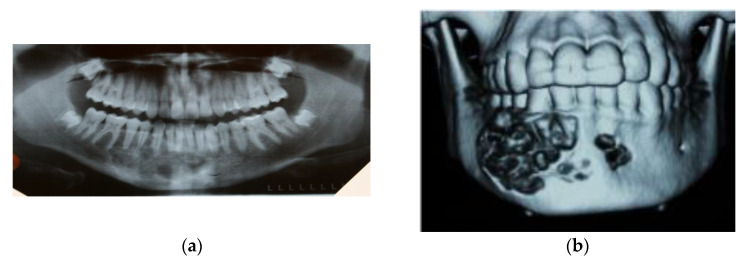
Imaging of the lesion: (**a**) an orthopantomographic radiograph showing a multi-cavity osteolytic lesion; (**b**) a CT scan of patient’s mandible (3d reconstruction)—a visible osteolytic loss destructing the mandibular cortical layer.

**Figure 3 medicina-59-00250-f003:**
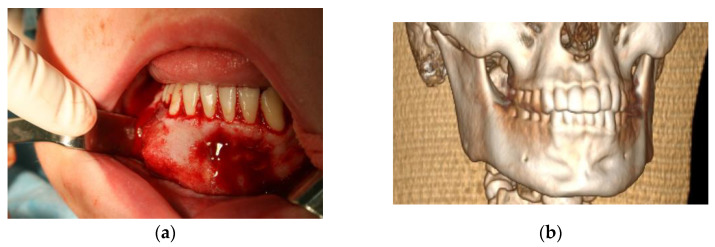
Clinical and radiological view after injections: (**a**) visible distension of the body of the mandible in the area of teeth 46–33—an intraoperative view after the elevation of the mucoperiosteal flap; (**b**) a CT scan of a patient’s mandible (3D reconstruction) bone of the body of the mandible after modeling by means of piezo surgery.

**Figure 4 medicina-59-00250-f004:**
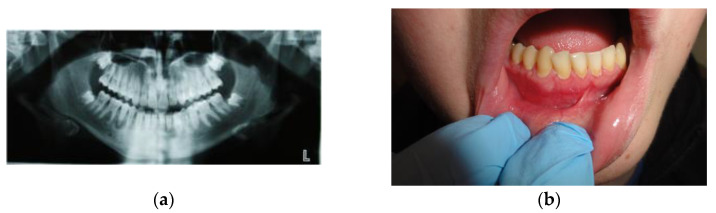
Clinical and radiological view after 6 months: (**a**) no radiological features of recurrence; (**b**) no clinical signs of recurrence.

**Figure 5 medicina-59-00250-f005:**
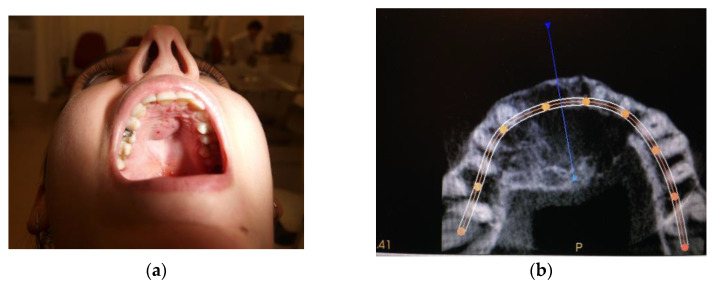
Clinical and radiological view before treatment: (**a**) palatal distension in the midline; (**b**) CT scan of a patient’s maxilla.

**Figure 6 medicina-59-00250-f006:**
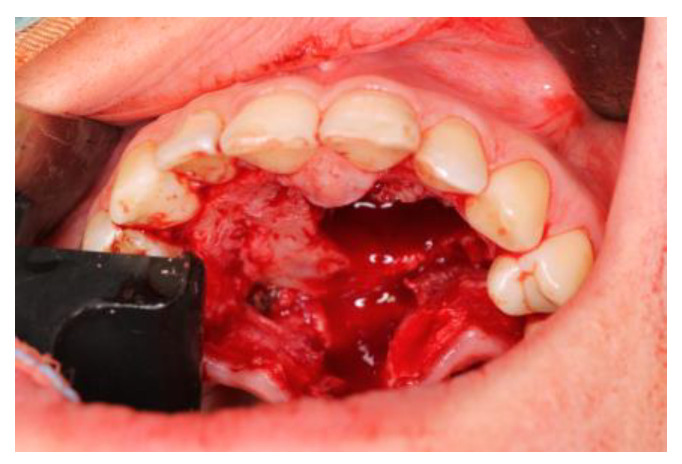
The bone of the maxilla during surgery—bone modeling by means of piezo surgery.

**Figure 7 medicina-59-00250-f007:**
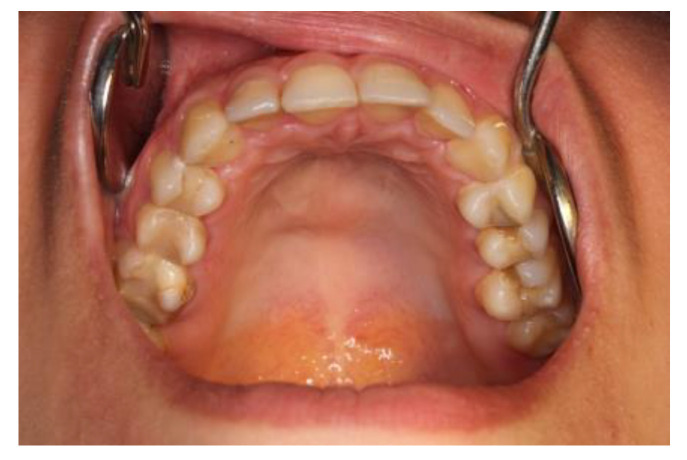
A view 4 months after finishing the therapy. No features of recurrence are detected clinically and radiologically.

**Table 1 medicina-59-00250-t001:** Summary of basic data for the case series.

Case #	Age	Sex	Injections’ Frequency and Doses of the Drug	Tumor Location	Final Histopathological Verification	Treatment Effects	Follow-Up
Case #1	14 y.o.	Male	Dexamethasone for 3 months 12 mg/3 mL every week and then 8 mg/12 mL every 2 weeks for the next 3 months	Mandible	Yes	Success	9 years
Case #2	29 y.o.	Female	Dexamethasone for 4 months 12 mg/3 mL every week and then 8 mg/12 mL every 2 weeks for the next 3 months	Maxilla	Yes	Success	8 years
Case #3	18 y.o.	Female	Dexamethasone for 3 months—84 mg/21 mL at 2-day intervals	Mandible	Yes	Surgical treatment was necessary	5 years

## Data Availability

Not applicable.

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
