# Peer review of "Bony Canal Method of Dexamethasone Injections in Aggressive Form of Central Giant Cell Granuloma—Case Series"

_medicina, 2023, doi:10.3390/medicina59020250_

Round 1

Reviewer 1 Report

Thank you for the interesting manuscript.

Please find below some grammatical points and feedback on areas that can be improved:

- Line 15: method replaced with methods

- Line 16: lead to the patient avoiding...

- Line 18: remove "the' after present

- Line 22: replace "one patient" with "The third patient"

- Line 25: replace "should" with "could be a ...". You can not be sure yet that injections "should be used" as you have only reported 2 cases. More clinical trials could be required.

- Line 33: reduce space after "is"

- Line 41: replace "and" with "with"

- Line 56: add something like "is observed" after the word septa.

- Line 58: may suggest "a" different ..

- Line 101: do you mean "complicate their facial/skeletal development"?

- Line 104: what is the term "novel" referring to? you need to emphasize, is it the bony canals? or the use of dexamethasone? The latter has been reported in case studies in the literature, see below examples:

Bayar OF, Ak G. Treatment of giant cell granuloma with intralesional corticosteroid injections: a case report. J Istanb Univ Fac Dent. 2015 Oct 21;49(3):45-50. doi: 10.17096/jiufd.88120. PMID: 28955545; PMCID: PMC5573504.

El Hadidi YN, Ghanem AA, Helmy I. Injection of steroids intralesional in central giant cell granuloma cases (giant cell tumor): Is it free of systemic complications or not? A case report. Int J Surg Case Rep. 2015;8C:166-70. doi: 10.1016/j.ijscr.2015.02.001. Epub 2015 Feb 7. PMID: 25699662; PMCID: PMC4353960.

- Line 115: replace "the patients" with "three patients"

- Line 132: start with "An" rather than "A"

- Line 132: "Ortopantomographic" you mean "Orthopantomographic"? it should be ortho not orto. Please check throughout manuscript.

- Line 138: perhaps describe more the bony canal features

- Line 150: 9-years

- Line 162: check space after Dexaven

- Line 169: replace "there were found a normal cancellous bone" with "normal cancellous bone was identified".

- Line 273: not sure it could be concluded that "Dexamethasone injections should be the recommended treatment" based on a few clinical cases. It rather "could be the recommended treatment".

- Perhaps you can recommend further investigation/research into this, as the more (suitable) cases treated, the better/bigger picture is obtained.

- Line 355: please check beginning of first reference. 

Best wishes

Author Response

Dear Reviewers,

Thank you for giving us the opportunity to submit a revised manuscript. We appreciate the time and effort that you dedicated to providing feedback on our manuscript and are grateful for the insightful comments on and valuable improvements to our paper. We have incorporated most of the suggestions to the revised manuscript. Those changes are highlighted within the manuscript. Please see below for a point-by-point response to your comments and concerns.

Reviewer 1:

1. We are very sorry for these mistakes - we have corrected the highlighted grammatical errors and template-based layout.

2. line 25 and 273: we have substituted the words should for could

3. line 104: we have described that term novel was reffered to bony canals method of dexamethasone injections not described in the available literature.

4. line 138: we have described bony canals method

Thank you very much for your time and comments so that we can improve the article

Best regards

Dear Reviewers,

Thank you for giving us the opportunity to submit a revised manuscript. We appreciate the time and effort that you dedicated to providing feedback on our manuscript and are grateful for the insightful comments on and valuable improvements to our paper. We have incorporated most of the suggestions to the revised manuscript. Those changes are highlighted within the manuscript. Please see below for a point-by-point response to your comments and concerns.

Reviewer 2:

1. line 1: We have changed article type to case report and add case series to article title.

2. we have corrected the highlighted grammatical errors and template-based layout.

3. We have removed section methods

4. We have structured section – case series into 3 subsections

5. We have checked the article according to the CARE guidelines

Thank you very much for your time and comments so that we can improve the article

Best regards

Mateusz Bielecki

Reviewer 2 Report

The authors conducted a paper in which they present three cases of aggressive forms of CGCG of the jaw treated with dexamethasone injections into the tumor mass.

There are several issues with this manuscript that the authors should address:

-Change article type to "case report" - line 1

-Include the article type ("case series") in the title.

-For case series articles there is no need for the methods section

-I consider that you should structure the section "3. Case series" in several subsections (example: 3.1. Case 1; 3.2. Case 2...)

- Structure your case presentation according to the CARE guidelines (https://www.care-statement.org/checklist). 

Author Response

Dear Reviewers,

Thank you for giving us the opportunity to submit a revised manuscript. We appreciate the time and effort that you dedicated to providing feedback on our manuscript and are grateful for the insightful comments on and valuable improvements to our paper. We have incorporated most of the suggestions to the revised manuscript. Those changes are highlighted within the manuscript. Please see below for a point-by-point response to your comments and concerns.

Reviewer 2:

1. line 1: We have changed article type to case report and add case series to article title.

2. we have corrected the highlighted grammatical errors and template-based layout.

3. We have removed section methods

4. We have structured section – case series into 3 subsections

5. We have checked the article according to the CARE guidelines

Thank you very much for your time and comments so that we can improve the article

Best regards

Mateusz Bielecki